# Invertible Convolutional Flow

**Mahdi Karami**[*]
[*]Department of Computer Science
University of Alberta
karami1@ualberta.ca

**Jascha Sohl-Dickstein**[†]    **Dale Schuurmans**[†][*]    **Laurent Dinh**[†]    **Daniel Duckworth**[†]
[†]Google Brain

## Abstract

Normalizing flows can be used to construct high quality generative probabilistic models, but training and sample generation require repeated evaluation of Jacobian determinants and function inverses. To make such computations feasible, current approaches employ highly constrained architectures that produce diagonal, triangular, or low rank Jacobian matrices. As an alternative, we investigate a set of novel normalizing flows based on the circular and symmetric convolutions. We show that these transforms admit efficient Jacobian determinant computation and inverse mapping (deconvolution) in $\mathcal{O}(N \log N)$ time. Additionally, element-wise multiplication, widely used in normalizing flow architectures, can be combined with these transforms to increase modeling flexibility. We further propose an analytic approach to designing nonlinear elementwise bijectors that induce special properties in the intermediate layers, by implicitly introducing specific regularizers in the loss. We show that these transforms allow more effective normalizing flow models to be developed for generative image models.

## 1   Introduction

Flow-based generative networks have shown tremendous promise for modeling complex observations in high dimensional datasets. In flow-based models, a complex probability density is constructed by transforming a simple base density, such as a standard normal distribution, via a chain of smooth, invertible mappings (bijections), to yield a *normalizing flow*. Such models are employed in various contexts, including approximating a complex posterior distribution in variational inference [Rezende and Mohamed, 2015], or for density estimation with generative models [Dinh et al., 2016].

Using a complex transformation (bijective function) to define a normalized density requires the computation of a Jacobian determinant, which is generally impractical for arbitrary neural network transformations. To overcome this difficulty and enable fast computation, previous work has carefully designed architectures that produce simple Jacobian forms. For example, [Rezende and Mohamed, 2015, Berg et al., 2018] consider transformations with a Jacobian that corresponds to low rank perturbations of a diagonal matrix, enabling the use of Sylvester's determinant lemma. Other works, such as [Dinh et al., 2014, 2016, Kingma et al., 2016, Papamakarios et al., 2017], use a constrained transformation where the Jacobian has a triangular structure. The latter approach has proved particularly successful, since this constraint is easy to enforce without major sacrifices in expressiveness or computational efficiency. More recently, Kingma and Dhariwal [2018] propose the use of $1 \times 1$ convolutions for cross channel mixing in a multi-channel signal, achieving tractability via a block diagonal Jacobian. Nevertheless, these models have overlooked some opportunities for formulating tractable normalizing flows that can enhance expressiveness and better capture the structure of natural data, such as images and audio. Also, a new line of work based on ordinary

differential equations has emerged recently that offers promising continuous dynamics based flows [Grathwohl et al., 2019].

In this work, we propose an alternative nonlinear convolution layer, the *nonlinear adaptive convolution filter*, where expressiveness is increased by allowing a layer's kernel to adapt to the layer's input. The idea is to partition the input of a layer $\boldsymbol{x}$ into $\{\boldsymbol{x}_1, \boldsymbol{x}_2\}$, where the convolution updates $\boldsymbol{x}_2$ as $\boldsymbol{w}(\boldsymbol{x}_1)*\boldsymbol{x}_2$, while the kernel $\boldsymbol{w}(\boldsymbol{x}_1)$ is a function of $\boldsymbol{x}_1$ that can be expressed by a deep neural network. We present invertible convolution operators whose Jacobian can be computed efficiently, making this approach practical for normalizing flow. Unlike the causal convolution employed in [van den Oord et al., 2016] to generate audio waveforms, or in [Zheng et al., 2017] to approximate the posterior in a variational autoencoder, the proposed transformations are not constrained to depend only on the preceding input variables and also offer efficient inverse mapping, also known as deconvolution, analytically. Also, recently, circular convolution has been adopted in [Karami et al., 2018] as a normalizing flow for density estimation and in [Hoogeboom et al., 2019] to design invertible periodic convolution for (almost) periodic data. Furthermore, we propose an analytic approach to add invertible pointwise nonlinearity in the flow that implicitly induces specific regularizers on the intermediate layers.

## 2 Background

Given a random variable $\mathbf{z} \sim p(\mathbf{z})$ and an invertible and differentiable mapping $g : \mathbb{R}^n \to \mathbb{R}^n$, with inverse mapping $f = g^{-1}$, the probability density function of the transformed variable $\boldsymbol{x} = g(\mathbf{z})$ can be recovered by the *change of variable rule* as $p(\boldsymbol{x}) = p(\mathbf{z}) \left|\det \boldsymbol{J}_g\right|^{-1} = p(f(\boldsymbol{x})) \left|\det \boldsymbol{J}_f\right|$. Here $\boldsymbol{J}_g = \frac{\partial g}{\partial \mathbf{z}^\top}$ and $\boldsymbol{J}_f = \frac{\partial f}{\partial \boldsymbol{x}^\top}$ are the Jacobian matrices of functions $g$ and $f$, respectively. One can use these to build a complex mapping $g$ by composing a chain of simple bijective maps, $g = g^{(1)} \circ g^{(2)} \circ ... \circ g^{(K)}$, that preserve invertibility, with the inverse mapping being $f = f^{(K)} \circ f^{(K-1)} \circ ... \circ f^{(1)}$. By applying the chain rule to the Jacobian of the composition, and using the fact that $\det \boldsymbol{AB} = \det \boldsymbol{A} \det \boldsymbol{B}$, the log-likelihood equality (LLE) can be written as

$$\log p(\boldsymbol{x}) = \log p(\mathbf{z}) + \sum_{k=1}^{K} \log \left|\det \boldsymbol{J}_{f_k}\right|. \tag{1}$$

Evaluating the Jacobian determinant is the main computational bottleneck in (1) since, in general, its scaling is cubic in the size of input. It is therefore natural to seek structured transformations that mitigate this cost while retaining useful modeling flexibility.[1]

### 2.1 Toeplitz structure and Circular Convolution

Although available methods have typically considered bijections whose Jacobians have block-diagonal or triangular forms, these are not the only useful possibilities. In fact, various other transformations exist whose Jacobian has sufficient structure to allow computationally efficient determinant calculation. One such structure is the *Toeplitz* property, where all the elements along each diagonal of a square matrix are identical (Figure 1(a)). The calculation of the determinant can then be simplified significantly. Let $\boldsymbol{J}_T$ be a Toeplitz matrix of size $N \times N$; its determinant can be evaluated in $\mathcal{O}(N^2)$ time in general [Monahan, 2011]. More specifically, if $\boldsymbol{J}_T$ has a limited bandwidth size of $K = r + s$, as depicted in Figure 1(a), then the determinant computation can be reduced to $\mathcal{O}(K^2 \log N + K^3)$ time [Cinkir, 2011]. Moreover, Toeplitz matrices can be inverted efficiently [Martinsson et al., 2005]. The fact that the discrete convolution can be expressed as a product of a Toeplitz matrix and the input [Gray et al., 2006] highlights that the Toeplitz property is of particular interest in *convolutional neural networks (CNNs)*.

$$\mathbf{J}_T = \begin{bmatrix} w_0 & w_{-1} & \dots & w_{-s} & & \mathbf{0} \\ w_1 & w_0 & & & & \\ \vdots & \ddots & \ddots & \ddots & \ddots & \\ w_r & \ddots & \ddots & \ddots & \ddots & w_{-s} \\ & \ddots & \ddots & \ddots & w_0 & w_{-1} \\ \mathbf{0} & & w_r & \dots & w_1 & w_0 \end{bmatrix}$$

(a)

$$\mathbf{J}_C = \begin{bmatrix} w_0 & w_{N-1} & \dots & w_2 & w_1 \\ w_1 & w_0 & \ddots & \ddots & w_2 \\ \vdots & \ddots & \ddots & \ddots & \vdots \\ w_{N-2} & \ddots & \ddots & w_0 & w_{N-1} \\ w_{N-1} & w_{N-2} & \dots & w_1 & w_0 \end{bmatrix}$$

(b)

$$\mathbf{J}_S = \begin{bmatrix} w_0 & w_0 & \dots & w_{N-3} & w_{N-2} \\ w_1 & w_0 & \ddots & w_{N-4} & w_{N-3} \\ \vdots & \ddots & \ddots & \ddots & \vdots \\ w_{N-2} & w_{N-3} & \ddots & w_0 & w_0 \\ w_{N-1} & w_{N-2} & \dots & w_1 & w_0 \end{bmatrix} + \begin{bmatrix} w_1 & w_2 & \dots & w_{N-1} & w_{N-1} \\ w_2 & w_3 & \reflectbox{$\ddots$} & w_{N-1} & w_{N-2} \\ \vdots & \reflectbox{$\ddots$} & \reflectbox{$\ddots$} & \reflectbox{$\ddots$} & \vdots \\ w_{N-1} & w_{N-1} & \reflectbox{$\ddots$} & w_2 & w_1 \\ w_{N-1} & w_{N-2} & \dots & w_1 & w_0 \end{bmatrix}$$

(c)

Figure 1: (a) $\mathbf{J}_T$ is a Toeplitz matrix with limited bandwidth size of $K = r + s$, (b) $\mathbf{J}_C$ is the Jacobian of circular convolution that is a circulant matrix, and (c) $\mathbf{J}_S$ is the Jacobian of symmetric convolution that can be expressed as summation of a Toeplitz matrix and an upside-down Toeplitz matrix (also called a Hankel matrix where its skew-diagonal elements are identical).

In this paper, we consider a particular transformation whose Jacobian is a *circulant matrix*, a special form of Toeplitz structure where the rows (columns) are cyclic permutations of the first row (column), *i.e.* $J_{l,m} = J_{1,(l-m) \bmod N}$. See Figure 1(b) for an illustration. This structure allows certain computationally expensive algebraic operations, such as determinant calculation, inversion and eigenvalue decomposition, to be performed efficiently in $\mathcal{O}(N \log N)$ time by exploiting the fact that a square circulant matrix can be diagonalized by a discrete Fourier transform (DFT) [Gray et al., 2006]. Define the circular convolution as $\boldsymbol{y} := \boldsymbol{w} \circledast \boldsymbol{x}$ where $\boldsymbol{y}(i) := \sum_{n=0}^{N-1} \boldsymbol{x}(n)\boldsymbol{w}(i - n)_{\bmod N}$, which is equivalent to the linear convolution of two sequences when one is padded cyclically, also known as periodic padding, as illustrated in Figure 2(a). The key property we exploit in developing an efficient normalizing layer is that the Jacobian of this convolution forms a circulant matrix, hence its determinant and inverse mapping (deconvolution) can be computed efficiently. Some useful properties of this operation are needed:

**Proposition 1** *Let $\boldsymbol{y} := \boldsymbol{w} \circledast \boldsymbol{x}$ be a circular convolution on the input vector $\boldsymbol{x}$ with its DFT transform $\boldsymbol{x}_{\mathcal{F}} := \mathcal{F}_{DFT}\{\boldsymbol{x}\}$. Then:*

*a) The circular convolution operation can be expressed as a vector-matrix multiplication $\boldsymbol{y} = \boldsymbol{C}_w \boldsymbol{x}$ where $\boldsymbol{C}_w$ is a circulant square matrix having the convolution kernel $\boldsymbol{w}$ as its first row.*

*b) The Jacobian of the mapping is $\boldsymbol{J}_y = \boldsymbol{C}_w$.*

*c) The matrix $\boldsymbol{C}_w$ can be diagonalized using DFT basis with its eigenvalues being equal to the DFT of $\boldsymbol{w}$, hence $\log |\det \boldsymbol{J}_y| = \sum_{n=0}^{N-1} \log |\boldsymbol{w}_{\mathcal{F}}(n)|$.*

*d) The circular convolution can be expressed by element-wise multiplication in the frequency domain, $\boldsymbol{y}_{\mathcal{F}}(k) = \boldsymbol{w}_{\mathcal{F}}(k)\,\boldsymbol{x}_{\mathcal{F}}(k)$, a.k.a. the circular convolution-multiplication property.*

*e) If $\boldsymbol{w}_{\mathcal{F}}(n) \neq 0 \;\forall n$, this linear operation is invertible with inverse $\boldsymbol{x}_{\mathcal{F}}(n) = \boldsymbol{w}_{\mathcal{F}}^{-1}(n)\,\boldsymbol{y}_{\mathcal{F}}(n)$. Moreover, its inverse mapping (deconvolution) is also a circular convolution operation with kernel $\boldsymbol{w}^{inv} := \mathcal{F}_N^{-1}\{\boldsymbol{w}_{\mathcal{F}}^{-1}\}$. On the other hand, the log determinant Jacobian also acts as a log-barrier in the objective function that in turn prevents the $\boldsymbol{w}_{\mathcal{F}}(n)$ from becoming zero hence enforces the invertibility of the convolution filter.*

*f) The circular convolution, its inverse, and Jacobian determinant can all be efficiently computed in $\mathcal{O}(N \log N)$ time in the frequency domain, exploiting Fast Fourier Transform (FFT) algorithms.*

## 2.2 Symmetric convolution

Circular convolution is not a unique operation with such properties, *symmetric convolution* is another form of structured filtering operation that can be adopted to achieve interesting desirable properties.

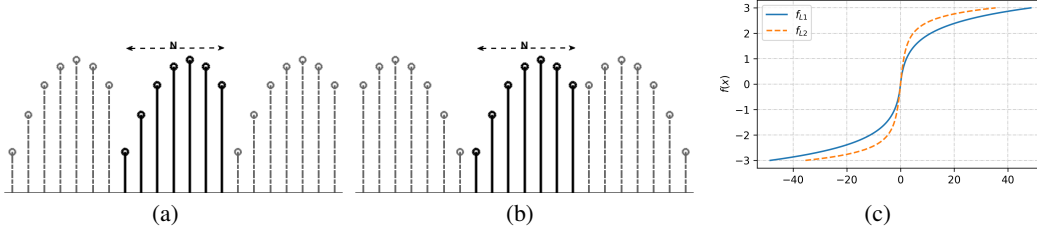

Figure 2: (a) Cyclic (periodic) extension and (b) even-symmetric extension of the base sequence, where the base sequence specified by dark solid lines. (c) Nonlinear gates corresponding to $l1$ and $l2$ regularizers.

A family of symmetric extension (padding) patterns and their corresponding discrete trigonometric transforms (DTT) are outlined in Martucci [1994], based on which alternative symmetric convolution filters can be defined that satisfy the convolution-multiplication property. Among this family, we choose an even-symmetric extension that can be readily interpreted. Define an even-symmetric extension of a base sequence of length $N$ around $N - 1/2$ as

$$\hat{\boldsymbol{x}}(n) = \varepsilon\{\boldsymbol{x}(n)\} := \begin{cases} \boldsymbol{x}(n) & n = 0, 1, ..., N-1 \\ \boldsymbol{x}(-n-1) & n = -N, ..., -1 \end{cases}. \qquad (2)$$

This even-symmetric extension is illustrated in Figure 2(b). The *symmetric convolution* of two sequences, denoted by $*_s$, can then be defined by the circular convolution of their corresponding even-symmetric extensions, as $\boldsymbol{y} = \boldsymbol{w} *_s \boldsymbol{x} := \mathcal{R}\{\hat{\boldsymbol{x}} \circledast \hat{\boldsymbol{w}}\}$, where $\mathcal{R}\{.\}$ is a rectangular window operation that retains the base sequence of interest in an extended sequence; that is, it inverts the symmetric extension operation (2). Now, since the sequences are extended by an even-symmetric pattern, the cosine functions provide the appropriate basis for the Fourier transform, giving rise to the discrete cosine transform of type two (DCT-II):

$$\boldsymbol{x}_{\mathcal{C}}(k) = \mathcal{F}_{dct}\{\boldsymbol{x}\}_k = \frac{1}{\sqrt{N}} \sum_{n=0}^{N-1} \frac{\sqrt{2}}{\sqrt{1_{n=0}+1}} \boldsymbol{x}(n) \cos\left(\frac{\pi k}{N}\left(n + \tfrac{1}{2}\right)\right). \qquad (3)$$

The convolution-multiplication property holds for this convolution, which implies that the symmetric convolution of two sequences in the spatial domain can be expressed as a pointwise multiplication in the transform domain, after a forward DCT of its operands, *i.e.* $\boldsymbol{y}_{\mathcal{C}} = \boldsymbol{w}_{\mathcal{C}} \odot \boldsymbol{x}_{\mathcal{C}}$. This property also offers and alternative definition for the symmetric convolution: the inverse DCT of pointwise multiplication of the forward DCT of its operands [Martucci, 1994].

One can also show that the symmetric convolution provides a structured Jacobian that can be specified by Toeplitz matrices; see Figure 1(c) for an illustration. Analogous to the results presented in Proposition 1 for circular convolution, the symmetric convolution-multiplication property implies that the Jacobian of the symmetric convolution can be diagonalized by a DCT basis, with eigenvalues being the DCT of the convolution kernel. Similarly, the inverse filter (*deconvolution*) can be obtained by inverting the kernel coefficients in the transform domain, i.e. $\boldsymbol{w}^{inv} := \mathcal{F}_{dct}^{-1}\{1./\boldsymbol{w}_{\mathcal{C}}\}$, where, again, the *invertibility* of the convolution is guaranteed by the fact that it log determinant Jacobian in the objective function keeps the elements of $\boldsymbol{w}_{\mathcal{C}}$ away from zero (as a log-barrier). On the other hand, since the DCT can be defined in terms of a DFT of the symmetric extension of the original sequences, the symmetric convolution, its inverse, and Jacobian determinant can exploit available fast Fourier algorithms with $\mathcal{O}(N \log N)$ complexity.[2]

## 3 Convolutional normalizing flow

### 3.1 Data adaptive convolution layer

The special convolutional forms introduced above appear to be particularly well suited to capturing structure in images and audio signals, therefore we seek to design more expressive normalizing flows using the convolution bijections as a building blocks. To increase flexibility, we propose a *data-adaptive convolution* filter with a filter kernel that is a function of the input of the layer.

Inspired by the idea of the coupling layer in [Dinh et al., 2016], a modular bijection can be formed by splitting the input $\boldsymbol{x} \in \mathbb{R}^d$ into two disjoint parts $\{\boldsymbol{x}_1 \in \mathbb{R}^{d_1}, \boldsymbol{x}_2 \in \mathbb{R}^{d_2} : d_1 + d_2 = d\}$, referred to as the *base input* and *update input*, respectively, and only updating $\boldsymbol{x}_2$ by an invertible convolution operation with a data-parameterized kernel that depends on $\boldsymbol{x}_1$. The data-adaptive convolution sub-flow can then be expressed as

$$f_*(\boldsymbol{x}_2; \boldsymbol{x}_1) = \boldsymbol{w}(\boldsymbol{x}_1) * \boldsymbol{x}_2. \tag{4}$$

In the above transformation $*$ is an invertible convolution operation and can be one of the invertible convolutions introduced in last section. Here, the kernel $\boldsymbol{w}(\boldsymbol{x}_1)$ can be any nonlinear function, which leads to a *nonlinear adaptive convolution* filtering scheme.

## 3.2 Pointwise nonlinear bijections

Adding pointwise nonlinear bijections in the chain of normalizing flows can further enhance expressiveness. More specifically, focusing on the Jacobian determinant introduced by the nonlinearities in log-likelihood equation (1), one can observe that these terms can be interpreted as regularizers on the latent representation. In other words, specific structures on intermediate activations can be encouraged by designing customized pointwise nonlinear gates; these structures encode various prior knowledge into the design of the model. Let $\sigma^{(k)}$ denote the $k^{th}$ bijection in the chain of normalizing flows that is assumed to be an pointwise nonlinear operation, *i.e.* $\boldsymbol{y}_i^{(k)} = \sigma^{(k)}(\boldsymbol{x}_i^{(k)})$. Dropping the indices, this mapping can be simply written as $y = \sigma(x)$ with inverse $x = \phi(y) = \sigma^{-1}(y)$. Since the nonlinearity operates elementwise, its Jacobian is diagonal, hence the log determinant reduces to $\log|\det \boldsymbol{J}_y| = \sum_{i=1}^{d} \log\left|\frac{\partial \sigma(x_i)}{\partial x_i}\right|$. Then, an analytic approach designing nonlinear invertible gates are derived in the following.

**Proposition 2** *Assume we want to induce a specific structure, formulated by a regularizer $\gamma(y)$, on the intermediate activation $y := \boldsymbol{y}_i^{(k)}$. Then the elementwise bijection can be defined as the solution to the differential equation: $|\frac{\partial \sigma^{-1}}{\partial y}| = |\frac{\partial \phi}{\partial y}| = e^{\gamma(y)}$. In the other word, the contribution to the $-\log|\det \boldsymbol{J}_\sigma|$ term in the negative log-likelihood from this unit will then reduces to $\log|\frac{\partial \phi}{\partial y}| = \gamma(y)$.*

Solving the above equation and deriving the nonlinear bijection for two well established $l1$ and $l2$ regularizers leads to the following.

- $l1$ regularization: $\gamma(y) = \alpha|y|$ which corresponds to Laplace distribution assumption on $y$:

$$\phi_\alpha(y) = \frac{\text{sign}(y)}{\alpha}(e^{\alpha|y|} - 1), \quad \sigma_\alpha(x) = \frac{\text{sign}(x)}{\alpha}\ln(\alpha|x| + 1). \tag{5}$$

  Due to its symmetric logarithmic shape, we call the forward function $\sigma_\alpha(x)$ an *S-Log* gate parameterized by positive-valued $\alpha$.

- $l2$ regularization: $\gamma(y) = \alpha y^2$ which corresponds to Gaussian distribution assumption on $y$:

$$\phi_\alpha(y) = \sqrt{\frac{\pi}{4\alpha}}\text{erfi}(\sqrt{\alpha}y), \quad \sigma_\alpha(x) = \frac{1}{\sqrt{\alpha}}\text{erfi}^{-1}(\sqrt{\frac{4\alpha}{\pi}}x).$$

The proposed nonlinear gates, plotted in Figure 2(c), are not only differentiable by construction but also have unbounded domain and range, making them suitable choices for designing normalizing flows in many settings such as density estimation. Due to its simple analytical form and closed form inversion, the *S-Log* gate, (5), is adopted as nonlinear bijection in our model architecture. For multichannel inputs, we assume that the gates share the same parameter $\alpha$ over all spatial locations of a channel (feature map).

## 3.3 Combined convolution multiplication layer

The convolution operation spatially slides a filter and applies the same weighted summation at every location of its input, resulting in location invariant filtering. To achieve a more flexible and richer filtering scheme, we can combine an element-wise multiplication, indicated by $f_\odot$, and invertible convolution, indicated by $f_*$, so that the filtering scheme varies over space and frequency. The product of a diagonal matrix with a circulant matrix was also proposed in [Cheng et al., 2015] as a

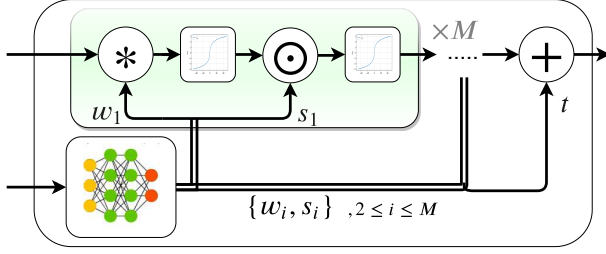

Figure 3: The diagram of one step of flow (CONF) that is composed of $M$ combined convolutional flows defined in (6). In density estimation, the input to the conditioning neural network is the base input, $x_1$, and the flow updates $x_2$. In variational inference applications, the neural network is conditioned on the data points $x$ while warping the latent random variable $z$.

structured approximation for dense (fully connected) linear layers, while [Moczulski et al., 2015] showed that any $N \times N$ linear operator can be approximated to arbitrary precision by composing order $N$ of such products.

Overall, the aforementioned components can be deployed to compose a *combined convolutional flow* as

$$f_{w,s}(x_2; x_1) = (\sigma_{\alpha'} \circ f_\odot \circ \sigma_\alpha \circ f_*)(x_2; x_1)$$
$$= \sigma_{\alpha'}\left( s(x_1) \odot \sigma_\alpha(w(x_1) * x_2) \right) \tag{6}$$

We found that a more expressive network can be achieved by stacking $M$ iterates of the combined convolutional flows and an additive coupling transform in each step of the network. Therefore, the *convolutional coupling flow* (*CONF*) can be written as

$$\begin{cases} y_1 = x_1 \\ y_2 = (f_{w,s}^{(M)} \circ ... \circ f_{w,s}^{(1)})(x_2; x_1) + t(x_1). \end{cases} \tag{7}$$

The parameters of the flow $\{w_1, s_1, ..., w_M, s_M, b\}$ can be any nonlinear function of the base input $x_1$ and are not required to be invertible, hence they can be modeled by deep neural networks with an arbitrary number of hidden units, offering flexibility and rich representation capacity while preserving an efficient learning algorithm. These are also called *conditioning networks* in the context of normalizing flow. The model complexity can be significantly reduced by using one conditioning neural network for all parameters of a coupling flow so that it shares all layers except the last one for generating the parameters of the flow. Consequently, we achieve a more expressive flow with the stack of bijectors in (7) without introducing too many extra NN layers in the model.

The modular structure of coupling CONF modules (7) implies that its Jacobian determinant can be expressed in terms of its sub-flows. More details on the Jacobian determinant, invertibility condition and inverse of this transformation can be found in Appendix A.

**Initialization of the parameters:** Better data propagation is expected to be achieved for very deep normalizing flows if the combined flow (6) acts (approximately) as an identity mapping at initialization. Accordingly, the parameters of the nonlinear bijector pair, $\{\sigma_\alpha, \sigma_{\alpha'}\}$, are initialized sufficiently close to zero so that they behave approximately as linear functions at the outset. Furthermore, the conditioning networks are initialized such that the scaling filters, $s$, and the convolution kernels at the frequency domain, $\mathcal{F}\{w\}$, are all initially identity filters.

**Multi-dimensional extension:** The multi-dimensional discrete Fourier transform can be expressed in separable forms, meaning that the operations can be performed by successively applying 1-dimensional transforms along each dimension [Gonzalez and Woods, 1992]. The separability property ensures the results mentioned so far can be extended to multi-dimensional settings. In this work, we are particularly interested in 2-D operations for image data. Based on the 2-D circular convolution definition, its equivalent block-circulant matrix form, and diagonalization method by 2-D DFT [Gonzalez and Woods, 1992, Ch. 5], the results of the circular convolution in Theorem 1 can be readily generalized to the 2-D case.[3] The same properties apply to the 2-D symmetric convolution, since the symmetric convolution-multiplication property can be generalized naturally to the 2-D setting [Foltz and Welsh, 1998].

Table 1: Average test negative log-likelihood (in nats) for tabular datasets and (in bits/dim) for MNIST and CIFAR using fully connected conditioning networks (lower is better). C-CONF and S-CONF stands for circular and symmetric convolutional coupling flow presented in (7), respectively. Error bars correspond to 2 standard deviations. The results of the benchmark methods are from Grathwohl et al. [2019].

| | POWER | GAS | BSDS300 | MNIST | CIFAR10 |
|---|---|---|---|---|---|
| MADE | $3.08 \pm .03$ | $-3.56 \pm .04$ | $-148.85 \pm .28$ | $2.04 \pm .01$ | $5.67 \pm .01$ |
| MAF | $-0.24 \pm .01$ | $-10.08 \pm .02$ | $-155.69 \pm .28$ | $1.89 \pm .01$ | $4.31 \pm .01$ |
| Real NVP | $-0.17 \pm .01$ | $-8.33 \pm .14$ | $-153.28 \pm 1.78$ | $1.93 \pm .01$ | $4.53 \pm .01$ |
| Glow | $-0.17 \pm .01$ | $-8.15 \pm .40$ | $-155.07 \pm .03$ | - | - |
| FFJORD | $-0.46 \pm .01$ | $-8.59 \pm .12$ | $-157.40 \pm .19$ | - | - |
| **S-CONF** | $\mathbf{-0.48} \pm .01$ | $\mathbf{-10.98} \pm .13$ | $\mathbf{-163.23} \pm .13$ | $\mathbf{1.26} \pm .01$ | $\mathbf{3.78} \pm .03$ |
| **C-CONF** | $-0.47 \pm .01$ | $-10.84 \pm .06$ | $\mathbf{-163.23} \pm .34$ | $\mathbf{1.25} \pm .01$ | $3.82 \pm .00$ |

Table 2: Results in bits per dimension for MNIST and CIFAR10 using CNN based conditioning networks. The results of the benchmark methods are from [Kingma and Dhariwal, 2018] and [Grathwohl et al., 2019]

| | Real NVP | Glow | FFJORD | S-CONF |
|---|---|---|---|---|
| MNIST | 1.06 | 1.05 | 0.99 | 1.00 |
| CIFAR10 | 3.49 | 3.35 | 3.40 | 3.34 |

## 4 Model architecture

A highly flexible and complex density approximation can be formed by composing a chain of the convolution coupling layers introduced in this work. As explained in Section 1, the determinant of the Jacobian and inverse of the composition can then be obtained readily. In addition to the invertible transformation introduced in this work, we use the following bijections in the final architecture of the normalizing flow.

**Cross-channel mapping (mixing)**   For multi-channel setting, the invertible convolution operation is performed in a depthwise fashion *i.e.* each input channel is filtered by a separate convolution kernel. Then cross channel information flow can be complemented by channel shuffling or using a $1 \times 1$ convolution. The latter offered significant improvement with small computational overhead in normalizing flows [Kingma and Dhariwal, 2018] hence, is applied after each convolutional coupling layer in our architecture. Also, for single channel inputs, assuming equal size splits $\{x_1, x_2\}$ (base input and update input), these can be treated as two separate channels of the input and the same technique can be applied to mix them after each coupling layer.

**Multiscale architecture**   To achieve latent representations at multiple scales and obtain more fine-grained features, a subset of latent variables can be factored out at the intermediate layers. This technique is very useful for large image datasets and can significantly reduce the computational cost in very deep models [Dinh et al., 2016].

**Normalization**   To improve the training in very deep normalizing flows, batch normalization was employed as a bijection after each coupling layer in [Dinh et al., 2016]. To overcome the adverse effect of small minibatch size in batch normalization, Kingma and Dhariwal [2018] proposed *actnorm*, as normalization, which applies an affine transformation and normalizes the activation per channel, similar to batch normalization but with larger minibatch size, at initialization while the parameters of this bijection are freely updated during training with smaller minibatch size, the technique called data dependent initialization. Thus, in density estimation experiments, we employed the actnorm layers as bijections in the chain of normalizing flow and also in the deep conditioning neural networks.

## 5 Experiments

### 5.1 Density estimation

We first conduct experiments to evaluate the benefits of the proposed flow model (CONF). As observed in [Huang et al., 2018], expressiveness of the affine coupling flows and affine autoregressive

flows stems from the complexity of the conditioning neural network that models flow parameters, and successive application of the flows. Therefore for fair comparison we follow [Papamakarios et al., 2017] and use a general-purpose neural network composed of fully connected layers in the design of conditioning networks. In this way we highlight the capacity of the flow itself, without relying on complex data dependent neural networks such as deep residual convolutional network used in [Dinh et al., 2016, Kingma and Dhariwal, 2018, Ho et al., 2019].

First we evaluate the proposed flow for density estimation on tabular datasets, considering two UCI datasets (POWR, GAS) and the natural image patches dataset (BSDS300) used in Papamakarios et al. [2017]. Description of these datasets and the preprocessing procedure applied can be found therein. We also perform unconditional density estimation on two image datasets; MNIST, consisting of handwritten digits [Y. LeCun, 1998] and CIFAR-10, consisting of natural images [Krizhevsky, 2009]. In BSDS300, the value of bottom-right pixel is replaced with the average of its immediate neighbors resulting in monochrome patches of size $8 \times 8$. For image data, the 2D invertible convolution is used as the flow. All datasets are dequantized by adding uniform distributed noise to each dimension, and then they are scaled to $[0, 1]$ values. Variational dequantization is proposed as a an alternative method offering better variational lower bound on the log-likelihood [Ho et al., 2019], which is beyond the scope of this paper.

We compare the density estimation performance of CONF to the affine coupling flow models real-NVP [Dinh et al., 2016] and Glow [Kingma and Dhariwal, 2018], and the recent continuous-time invertible generative model FFJORD [Grathwohl et al., 2019]. These reversible models admit efficient sampling with a single pass of the generative model. We also compare the density estimation capacity of the proposed model against the autoregressive based methods, MADE [Germain et al., 2015], MAF [Papamakarios et al., 2017]. These family of autoregressive normalizing flows require $\mathcal{O}(D)$ evaluations of the generative function to sample from the model, making them prohibitively expensive for high dimensional applications. The results, summarized in Table 1, highlight that the circular convolution-based (FFT-based) CONF (C-CONF) and symmetric convolution-based (DCT-based) CONF (S-CONF) offer significant performance gains over the other models. Since S-CONF outperforms C-CONF in most of the experiments, we use it as the main convolutional flow in the next experiments, simply referring to it as CONF. The significant performance improvement of CONF on image datasets suggest that the feedforward conditioning NN were able to capture 2D local structures.

To make a fair comparison, we used a feedforward neural network architecture similar to the one used for MAF [Papamakarios et al., 2017] except that we simplified the architecture by using a single network for all parameters of a flow layer, while MAF used separate networks for the scaling and shift parameters. Each coupling flow is composed of a maximum of $M = 2$ iterates of the combined convolution flow. The parameters of the network and number of layers are selected to be comparable to those used in [Papamakarios et al., 2017]. Details of model architecture and experimental setup together with more empirical results are presented in appendix.

## 5.2 Density estimation using CNN based conditioning networks

We further assess the performance of CONF when the conditioning networks are based on convolutional neural networks, which are specifically designed for image data. A shallow convolutional NN, similar to the one used in GLOW, is employed to generate the parameters of the flow, except that we use one NN to generate all the parameters of a layer, reducing the number of model parameters. The results of the experiments on MNIST and CIFAR10 data are presented in Table 2. The experimental setup and generated samples from the model can be found in Appendix C.1 and D, respectively.

## 5.3 Variational inference

We also evaluate the proposed normalizing flow as a flexible inference network for a variational auto-encoder (VAE) [Rezende and Mohamed, 2015]. Here flows are only conditioned on encoded data points, produced by the encoder, and transform the posterior distribution of the latent variable without a coupling connection, resulting in $\boldsymbol{z}^{(t)} = (f_{\boldsymbol{w},\boldsymbol{s}}^{(M)} \circ ... \circ f_{\boldsymbol{w},\boldsymbol{s}}^{(1)})(\boldsymbol{z}^{(t-1)}; \boldsymbol{x}) + \boldsymbol{t}(\boldsymbol{x})$. We compare the performance of the trained VAE using this convolutional flow against other approaches, including a non flow-based VAE with factorized Gaussian distributions, and flow-based VAE using inverse autoregressive flow (IAF), planar flow [Rezende and Mohamed, 2015, Kingma et al., 2016] and

Table 3: Average test negative log-likelihood (in nats) and negative evidence lower bound (ELBO) on four benchmark datasets (lower is better). Reported error bars correspond to 2 standard deviations calculated over 3 trials. The combination of number of flow steps $F$ and $M$ of each model is reported in the format (F-M).

| | MNIST -ELBO | MNIST NLL | Omniglot -ELBO | Omniglot NLL | Caltech Silhouettes -ELBO | Caltech Silhouettes NLL | Frey Faces -ELBO | Frey Faces NLL |
|---|---|---|---|---|---|---|---|---|
| VAE | $86.55 \pm .06$ | $82.14 \pm .07$ | $104.28 \pm .39$ | $97.25 \pm .23$ | $110.80 \pm .46$ | $99.62 \pm .74$ | $4.53 \pm .02$ | $4.40 \pm .03$ |
| IAF | $84.20 \pm .17$ | $80.79 \pm .12$ | $102.41 \pm .04$ | $96.08 \pm .16$ | $111.58 \pm .38$ | $99.92 \pm .30$ | $4.47 \pm .05$ | $4.38 \pm .04$ |
| Planar | $86.06 \pm .31$ | $81.91 \pm .22$ | $102.65 \pm .42$ | $96.04 \pm .28$ | $109.66 \pm .42$ | $98.53 \pm .68$ | $4.40 \pm .06$ | $4.31 \pm .06$ |
| **CONF(16-1)** | $83.89 \pm .03$ | $80.86 \pm .05$ | $98.35 \pm .27$ | $94.54 \pm .12$ | $108.64 \pm 1.71$ | $97.29 \pm .91$ | $4.43 \pm .01$ | $4.34 \pm .02$ |
| O-SNF(4-8) | $84.74$ | $81.04 \pm .15$ | $101.41 \pm .08$ | $95.25 \pm .09$ | $109.37 \pm .94$ | $97.78 \pm .47$ | $4.50 \pm .00$ | $4.39 \pm .01$ |
| **CONF(4-8)** | $83.22 \pm .05$ | $80.64 \pm .06$ | $97.17 \pm .08$ | $94.19 \pm .03$ | $104.09 \pm 1.03$ | $94.56 \pm .29$ | $4.41 \pm .01$ | $4.31 \pm .00$ |
| O-SNF(16-32) | $83.32 \pm .06$ | $80.22 \pm .03$ | $99.00 \pm .29$ | $93.82 \pm .21$ | $106.08 \pm .39$ | $94.61 \pm .83$ | $4.51 \pm .04$ | $4.39 \pm .05$ |
| **CONF(16-16)** | | | $96.35 \pm .05$ | $93.66 \pm .03$ | $101.10 \pm .49$ | $92.37 \pm .40$ | $4.39 \pm .02$ | $4.29 \pm .00$ |

Sylvester normalizing flows (SNF) as the building blocks of the normalizing flows. We used the encoder/decoder architecture of Berg et al. [2018] and the results of the available methods are adopted from this paper. The details of training procedure are summarized in Appendix C.2.

Although the proposed flow is slower than SNF of the same size, the results in Table 3 show that CONF outperforms Sylvester flow in most cases, and even smaller CONF models show similar or better capacity than larger SNF. Also, we observe that CONF with $M = 1$ outperforms planar flow by a wide margin on all datasets, except for FreyFaces which is a challenging dataset and prone to overfitting for large SNF; here large CONF ($F = 16, M = 16$) perform the best among all methods, so demonstrates less sensitivity to overfitting on the FreyFaces dataset.

**Number of parameters:** Let the stochastic latent variable be a $D$-dimensional vector $\boldsymbol{z} \in \mathbb{R}^D$ and the encoder's output be $\boldsymbol{e}(\boldsymbol{x}) \in \mathbb{R}^E$, then each step of CONF requires an additional $E \times (2MD + D) + 2M$ parameters to produce the flow parameters based on $\boldsymbol{e}(\boldsymbol{x})$, which is comparable to the number of parameters related to a step of planar flow if $M = 1$. This is of the same order of the number of parameters of Sylvester flow with a bottleneck of size $M$, which is $E \times (2MD + 2M^2 + M)$.

## 6 Conclusion

In this work we showed that circular and symmetric convolutions can be used as invertible transformations with fast and efficient inversion, deconvolution, and Jacobian determinant evaluation. These features make them well suited for designing flexible normalizing flows. Using these invertible convolutions, we introduced a family of data adaptive coupling layers, which consist of convolutions, where the kernel of the convolutions are themselves a function of the coupling layer input. We also analytically derived invertible pointwise nonlinearities that implicitly induce specific regularizers on intermediate activations in deep flow models. The results also helps better understand the role of nonlinear gates through the lens of their contribution to latent variables' distributions. Using these new architectural components, we achieved state of the art performance on several datasets for invertible normalizing flows with fast sampling.

## Footnotes

[1]**Notation definition:** Throughout the paper, invertible flows are denoted by $f$, while $f(\boldsymbol{x})$ is used for unconditional flows, and conditional (data-parameterized) flows are identified by $f(\boldsymbol{x}_2; \boldsymbol{x}_1)$ or $f(\boldsymbol{x}_2; \theta(\boldsymbol{x}_1))$ where the flow warps $\boldsymbol{x}_2$ conditioned on $\boldsymbol{x}_1$. Subscripts are intended to specify the type of flow or its parameters while superscripts enumerate the order of flows in the chain. For example, $f_*$ denotes the convolutional flow in general and $\sigma_\alpha$ is used to specify the pointwise nonlinear bijectors with its inverse being $\phi_\alpha$. Also, in general, $\boldsymbol{y}$ and $\boldsymbol{x}$ indicate the output and input of a flow, respectively and when referring to $k^{th}$ flow in the chain, we use $\boldsymbol{y}^{(k)}$ and $\boldsymbol{x}^{(k)}$ where $\boldsymbol{x}^{(k)} = \boldsymbol{y}^{(k-1)}$. Moreover, *circular convolution* and *symmetric convolution* are denoted by $\circledast$ and $*_s$, respectively, while $*$ denotes an invertible convolution in general, and $\boldsymbol{x}_\mathcal{F}$, $\boldsymbol{x}_\mathcal{C}$ and $\boldsymbol{x}_\mathcal{T}$ denote *DFT*, *DCT* and *trigonometric transform* of sample $\boldsymbol{x}$, respectively.

[2] All bijective convolutions in experiments were performed in transform domain using a fast Fourier transform algorithm.

[3] Due to the separability property, the 2-D DFT of matrices of size $N_1 \times N_2$ can be computed in $\mathcal{O}(N_1 N_2 (\log N_1 + \log N_2))$ time.

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
