[Supplementary Material · Symmetric_Convolutional_Flow_sup_new.pdf]

## A Jacobian determinant and inverse of coupling convoultional flow equation 6

Due to its modular structure, the Jacobian of (6) can be expressed in terms of the Jacobian of its sub-flow. More precisely, its Jacobian is

$$\boldsymbol{J}_y = \frac{\partial \boldsymbol{y}}{\partial \boldsymbol{x}^\top} = \begin{bmatrix} \boldsymbol{I}_{d_1} & \boldsymbol{0} \\ \frac{\partial \boldsymbol{y_2}}{\partial \boldsymbol{x}_1^\top} & \frac{\partial \boldsymbol{y_2}}{\partial \boldsymbol{x}_2^\top} \end{bmatrix}. \tag{7}$$

Noticeably, the Jacobian is a block triangular matrix, so its determinant can be readily computed as the product of determinant of the square diagonal blocks, therefore

$$\log |\det \boldsymbol{J}_y| = \sum_{i=1}^{M} \log \left| \det \boldsymbol{J}_{w,s}^{(i)} \right|$$

$$= \sum_{i=1}^{M} \log \left| \det \boldsymbol{J}_{g_{\alpha'}}^{(i)} \right| + \log \left| \det \boldsymbol{J}_\odot^{(i)} \right| + \log \left| \det \boldsymbol{J}_{f_\alpha}^{(i)} \right| + \log \left| \det \boldsymbol{J}_*^{(i)} \right| \tag{8}$$

where $\boldsymbol{J}_{w,s}^{(i)}$ denotes the Jacobian of $f_{w,s}^{(i)}$. According to the results presented for invertible convolutions in section 1, $\log \left| \det \boldsymbol{J}_*^{(i)} \right|$ can be computed efficiently in $\mathcal{O}(N \log N)$ times using the fast Fourier transform algorithm. Also, it is worth noting that this term plays the role of a log barrier in the final loss function that prevents the eigenvalues of the Jacobian from falling to zero hence guarantees the invertibility of the convolution transform. Then, the inverse transform of (6) is[5]

$$\begin{cases} \boldsymbol{x}_1 = \boldsymbol{y}_1 \\ \boldsymbol{x}_2 = (g_{\boldsymbol{w},\boldsymbol{s}}^{(1)} \circ ... \circ g_{\boldsymbol{w},\boldsymbol{s}}^{(M)})(\boldsymbol{y}_2 - \boldsymbol{t}(\boldsymbol{x}_1); \boldsymbol{x}_1) \end{cases}$$
$$\text{where } g_{\boldsymbol{w},\boldsymbol{s}}(\boldsymbol{y}_2; \boldsymbol{x}_1) = \boldsymbol{w}^{inv} * g_\alpha(\boldsymbol{s}^{inv} \odot g_{\alpha'}(\boldsymbol{y}_2).)$$

## B Ablations study

The coupling convolution flow (6) composed of two new components compared to the affine coupling flow, 1) the pointwise nonlinear bijector and 2) the data-adaptive convolution. In this ablation study, we asses the contribution of each of these components on the overall performance of the CONF. The results in Table 4 highlights the effect of each ablation relative to CONF. These results show that the nonlinear bijector, S-Log, contributes more than the data-adaptive convolution in the performance improvement of CONF, in this case study.

Table 4: Average validation negative log-likelihood (in nats) of the ablations on GAS dataset at 5600 epochs.

|  | CONF | ablation: linear gates | ablation: no convolution |
|---|---|---|---|
| GAS | -10.89 ± .13 | -10.12 ± .29 | -10.74 ± .06 |

## C Model architecture and training procedure

### C.1 Density estimation

To train the model, we used the Adam optimizer [Kingma and Ba, 2014] with initial learning rate of .001 which was decayed slowly to 0.0001 with exponentially decaying of rate .97. We apply `sigm()` to the output of conditioning network to obtain the scaling filters, $\boldsymbol{s}$ and the convolution kernels at the frequency domain, $\boldsymbol{w}_f$. Actnorm [Kingma and Dhariwal, 2018] is employed as normalization bijector in the chain of flow and as a layer in the NN. An $l2$ regularizer with coefficient of 5e-5 is applied on all the weights. Also to control overfitting, we use dropout layer with $p_{drop} = .2$ for MNIST. To transform MNIST data from a bounded to an unbounded domain, a logit mapping of the

form $y = \text{logit}(\alpha + (1-\alpha)\frac{x}{256})$ is applied with $\alpha = 10^{-6}$. All datasets are dequantized by adding uniform distributed noise to each dimension, and then they are scaled to $[0,1]$ values.

The aforementioned setting is used for both density estimation experiments in Table 1 and Table 2.

Normalizing flow architecture, NN architecture for parameter generation and other hyper parameters of the results reported in Table 1 are outlined in Table 5. Squeezing from space to channel dimension is applied Q times and followd by K flow step after each squeeze, that is showed in the format $Q \times K$ for MNIST and CIFAR10 in the Table. No factor out (splitting) is used. The squeeze and convolution together can be interpreted as dilated convolution of factor 2. Although, we used 2D invertible convolution flow for these two datasets but the general purpose fully connected feedforward conditioning NN is applied for parameter generation.

Table 5: Hyper parameters of the results reported in Table 1.

| Dataset | normalizing flow architecture | | NN architecture | | Minibatch size |
|---|---|---|---|---|---|
| | # flow steps | M (itertes per step) | # layers | # hidden units | |
| POWER | 10 | 2 | 2 | 200 | 10000 |
| GAS | 10 | 2 | 2 | 100 | 10000 |
| BSDS300 | 10 | 1 | 2 | 512 | 10000 |
| MNIST | 2×5 | 1 | 2 | 1024 | 512 |
| CIFAR10 | 3×4 | 2 | 2 | 1024 | 512 |

For the CNN based NN experiments of Table 2, the results of realNVP and GLOW on CIFAR10 dataset are adopted from Kingma and Dhariwal [2018]. GLOW uses multiscale architecture with 3 scales each one composed of 32 steps of flow and use different shallow neural networks with 2 hidden layers and 512 channels (width) for each parameter of the flow. Splitting is performed on the channels dimension only. After each scale a factor out with rate 1/2 is applied. We used the same architecture except that we use one NN to generate all parameters of a flow step but we doubled its width to 1024 channels. For MNIST, we again followed similar architecture for the normalizing flow where 2 scales each one composed of 12 steps of flow. The NN of depth 2 hidden layers with width of 512 channels are applied as the conditioning network. The results of realNVP and GLOW on MNIST dataset are adopted from Grathwohl et al. [2019] where they used the following flow structure:

$3 *$ (coupling layers with checkerboard masking) $+$ squeeze $+ 3 *$ (coupling layers with channel masking)$+$

$3 *$ (coupling layers with checkerboard masking) $+$ squeeze $+ 3 *$ (coupling layers with channel masking)$+$

$4 *$ (coupling layers with channel masking)

Each CONF is composed of $M = 2$ iterates of convolution-multiplication on both datasets.

## C.2 Variational inference

We employed the encoder/decoder architecture of Berg et al. [2018] with different optimization setting. We apply exp() to the output of encoder to obtain the scaling filters, $s$ and the convolution kernels at the frequency domain, $w_f$. Minibatch size of 500 samples (100 for FreyFaces) is selected and the other hyper parameters are adjusted according to get better training. The Adam optimizer [Kingma and Ba, 2014] is used for training with learning rate decaying from initial value $lr_{init}$ to $.1 \times lr_{init}$ after warmup.

The annealing, a.k.a. warm-up, procedure is used that gradually increase the effect of KL divergence term in the loss function Sønderby et al. [2016], but we found that, on FreyFaces dataset, our model train better without warm-up. The hyper-parameters are summarized in Table 6.

Table 6: Hyper parameters of VAE results reported in Table 3.

| Dataset | Minibatch size | # warmup | lr | $\epsilon_{Adam}$ |
|---|---|---|---|---|
| MNIST | 500 | 100 | 0.001 | 0.1 |
| Omniglot | 500 | 100 | 0.001 | 0.1 |
| FreyFaces | 100 | 0 | 0.0005 | 0.1 |
| Caltech | 500 | 2000 | 0.001 | 0.1 |

 # D Samples generated from the CONF model

(a)  (b)

Figure 3: Samples generated from an CONF model using CNN based conditioning NN that is trained on (a) the MNIST dataset and (b) the CIFAR-10 dataset.

(a)  (b)

Figure 4: Samples generated from an CONF model using general purpose fully connected NN as conditioning network that is trained on (a) the MNIST dataset and (b) the CIFAR-10 dataset.

Figure 5: Even-symmetric expansion around first and last element of the base sequence, where the base sequence specified by dark solid lines.

## E  Another symmetric convolution

There exist different extensions, here we define another type that can have straightforward interpretation. Let a base sequence be extended by an even-symmetric operation $\varepsilon\{.\}$ around its last element as

$$\hat{\boldsymbol{x}}(n) = \varepsilon\{\boldsymbol{x}(n)\} := \begin{cases} \boldsymbol{x}(n) & n = 0, 1, ..., N \\ \boldsymbol{x}(2N - n) & n = N + 1, ..., 2N - 1 \end{cases} \qquad (9)$$

this type of even-symmetric expansion is depicted in Figure 5. Again, the *symmetric convolution* of two sequences can be defined in terms of the circular convolution of their corresponding even-symmetric extensions as $\boldsymbol{y} = \boldsymbol{w} *_s \boldsymbol{x} = \mathcal{R}\{\hat{\boldsymbol{x}} \circledast \hat{\boldsymbol{w}}\}$ and also the convolution-multiplication property holds for this type given the discrete cosine transform defined as

$$\boldsymbol{x}_c(k) = \mathcal{F}_{dct}\{\boldsymbol{x}\}_k = \sum_{n=0}^{N} \boldsymbol{x}(n) \times 2\alpha_n \cos\left(\frac{\pi k n}{N}\right) \qquad (10)$$

$$\text{where } \alpha_n = \begin{cases} 1/2 & n = 0, N \\ 1 & otherwise \end{cases}$$

This is called DCT-I in the literature. It can be shown that the Jacobian matrix of this transform have the following structure

$$\boldsymbol{J}_S = \begin{bmatrix} w_0 & w_1 + w_1 & \dots & w_{N-2} + w_{N-2} & w_{N-1} \\ w_1 & w_0 + w_2 & \dots & w_{N-3} + w_{N-1} & w_{N-2} \\ \vdots & \vdots & & \vdots & \vdots \\ w_{N-2} & w_{N-3} + w_{N-1} & \dots & w_0 + w_2 & w_1 \\ w_{N-1} & w_{N-2} + w_{N-2} & \dots & w_1 + w_1 & w_0 \end{bmatrix}$$

Since scaling a column or row of a square matrix with factor $\alpha$, multiply its determinant by $\alpha$, hence the multiplying the first and last column of this matrix by factor of two give rise to

$$\boldsymbol{J}'_S = \begin{bmatrix} 2w_0 & w_1 + w_1 & \dots & w_{N-2} + w_{N-2} & 2w_{N-1} \\ 2w_1 & w_0 + w_2 & \dots & w_{N-3} + w_{N-1} & 2w_{N-2} \\ \vdots & \vdots & & \vdots & \vdots \\ 2w_{N-2} & w_{N-3} + w_{N-1} & \dots & w_0 + w_2 & 2w_1 \\ 2w_{N-1} & w_{N-2} + w_{N-2} & \dots & w_1 + w_1 & 2w_0 \end{bmatrix}$$

$$= \begin{bmatrix} w_0 & w_1 & \dots & w_{N-2} & w_{N-1} \\ w_1 & w_0 & \ddots & w_{N-3} & w_{N-2} \\ \vdots & \ddots & \ddots & \ddots & \vdots \\ w_{N-2} & w_{N-3} & \ddots & w_0 & w_1 \\ w_{N-1} & w_{N-2} & \dots & w_1 & w_0 \end{bmatrix} + \begin{bmatrix} w_0 & w_1 & \dots & w_{N-2} & w_{N-1} \\ w_1 & w_2 & \iddots & w_{N-1} & w_{N-2} \\ \vdots & \iddots & \iddots & \iddots & \vdots \\ w_{N-2} & w_{N-1} & \iddots & w_2 & w_1 \\ w_{N-1} & w_{N-2} & \dots & w_1 & w_0 \end{bmatrix}$$

where $\det(\boldsymbol{J}'_S) = 4\det(\boldsymbol{J}_S)$. Therefore, this symmetric convolution provides a structured Jacobian matrix that can be specified in terms of a Toeplitz matrix and an upside-down Toeplitz (also called a Hankel) matrix for determinant computation.

## Footnotes

[5]The inverse kernel $\boldsymbol{w}(\boldsymbol{y}_1)^{inv}$ can indeed be derived through the procedure explained in Theorem 1 for circular convolution or in a similar way for symmetric convolution.