[Reviews · NeurIPS 2019]

Reviewer 1



This submission presents two new and expressive types of normalizing flows where computation of Jacobian determinant is efficient. The main problem that I see is a rather linear presentation of the background material and the main theoretical contribution. The easy fix to the former would be to introduce a table that lists different normalizing flows, specifies any assumptions (e.g. volume preserving, diagonal Jacobian), lists computation costs, etc. The benefit of such simple addition would be immense. The easy fix to the latter would be to produce a diagram to illustrate what CONF is, how different sections are related to each element of it. Another problem that I see is that in a number of cases I see proofs by statement (...appears to be particularly well suited to capture structure in images and audio signals...) and procedural descriptions ("we do that, then we do that". E.g. for normalization, we use a data dependent initialization transform....). Each of these cases needs to be properly referenced and argued for by describing the problem, detailing possible solutions and then picking one that you argue is the best one to use. Overall, I believe that the submission is sufficiently original, lacks quality and clarity in some respects and is of sufficient significance to the specialists in normalizing flow community. Following authors feedback and other reviewers comments I hope that the authors will address the issues raised in an adequate fashion. Under this assumption I raised my score by 1.

Reviewer 2



I appreciated the answers to my questions in the author response regarding 2D convolutions and invertibility issues. I'd suggest being explicit about both of these things in the camera ready and especially about the 2D convolutions. I'd also like to note that I don't think all 2D convolutions are separable. Thus, it should be noted in the final version that your methods only handle separable 2D convolutions. ------ Original review ------ The paper presents a novel flow based on invertible convolutions. This primarily through an interesting synthesis of ideas from flows and linear algebra (specifically Toeplitz, circulant or block Toeplitz/circulant concepts). This could be form a foundational building block for future flow-based models. The clarity of some parts could be improved. Detailed comments to follow. 1. Details on 2D convolutions. The paper briefly mentions extensions to the 2D case at the end of section 3 but this definitely needs to be expanded. As one question, are the convolutions used in the experiments all separable convolutions (i.e. they can be computed by a 1D convolution along each dimension)? It is unclear from the paragraph if actual 2D convolutions (and related block Toeplitz matrices) are used in the experiments or merely mentioned as a future work possibility. If general 2D convolutions are used, then the details are definitely important here. The 1D case is relatively easy to understand but the 2D case seems a bit more complicated and helpful to the reader. Also, intuitions about the 2D case would be quite useful as well. I think I would be okay with a reduction in discussing the 1D case in order to develop the 2D case a bit more. 2. Clarity and/or intuitions regarding the point-wise nonlinearities connection to regularizers. It was not immediately clear that when mentioning "regularizers", the idea is regularizing the *latent representation* rather than regularizing the parameters directly. Is this correct? Usually, regularizers are used to explicitly operate on the parameters of the network. This might still be a reasonable term but I would make sure to clarify this. Also, note that using point-wise nonlinearities does not seem novel itself. Only possibly the interpretation of them as regularizing the latent representation seems novel. --- Minor comments --- How do you ensure that the convolutions are invertible? Some parameter settings for the kernel of the convolution will not be invertible since the corresponding Toeplitz matrix is singular. Maybe this is only enforced softly through the log determinant term but it would be great to at least mention this in the paper. Proposition 2 seems very wordy. Maybe split up into a few definitions and then the proposition rather than including everything into one. Overall, the proposition as it is currently written is very difficult to parse as a single idea. It seems that comparing against MADE and MAF is not quite fair because these do not ignore temporal or spatial structure and are more general than convolutional-based architectures. Maybe it's still reasonable to compare to them but it would be good to clarify why it makes sense to compare to these.

Reviewer 3



Originality: Whilst previous research may have been headed in this direction (perhaps Zheng et al. 2017), this paper has fleshed out those previous ideas fully and produced an excellent architecture. The paper is adequately cited. Quality: The submission is a complete piece of work, in that it clearly explains the background theory and achieves excellent empirical results. The authors take care to compare their model fairly to other normalising flows (e.g. controlling the network capacity to be the same as that used in other architectures when finding flow parameters). Clarity: The paper is well-written and easy to understand. Significance: I expect this architecture to become a useful tool for future ML research and applications.

[Author Response · NeurIPS 2019]

**Paper Title:**         **Invertible Convolutional Flow**

Thanks to all the reviewers for their time and helpful comments!

### Reviewer #1

Thank you for your valuable suggestions. We agree that including a table contrasting flow architectures, as you suggest, will greatly improve the presentation of past work. We are working to develop a clear diagram illustrating the CONF layer, and will include a diagram of this type in the camera ready. We will additionally reference and/or better motivate the design choices (eg, the use of ActNorm instead of batch normalization) for the camera ready.

### Reviewer #2

Thank you for your detailed comments.

*1. Details on 2D convolutions.*
The results presented for the 1D case are based on the convolution-multiplication property and the discrete trigonometric transforms (DFT or DCT) of the signal. Since the multi-dimensional transforms can be expressed separably in terms of 1D transforms, the theoretical results presented in this work can therefore be extended to 2D or 3D convolutions and their corresponding block circulant or block Toeplitz matrices. In practice, we used 2D invertible convolutions and 2D DFT/DCT for image datasets, which are implemented using the convolution-multiplication property and the efficient 1D FFT algorithm, thanks to their separable property. The explanation of 2D convolutions will be greatly expanded to clarify these points in the final version.

*2. Point-wise nonlinearities*
The nonlinear gates can induce special properties on the intermediate activations by introducing extra terms in the loss functions that, as you mentioned, can be interpreted as regularizers on the *latent representation*. Indeed, the main novelty of this part is proposing an analytic approach to designing customized pointwise nonlinearities according to desired latent structures in the deep normalizing flow. This also helps better understand the role of nonlinear gates through the lens of their contribution to latent variables' distributions. As you suggested, we will revise Proposition 2 to better clarify these ideas.

*3. Invertibilty of the convolutions*
The log determinant Jacobian of the convolutions acts as a log-barrier in the objective function that in turn prevents the convolution kernel in the frequency domain, $w_f(n)$, from becoming zero, and hence guarantees the invertibility of the convolution transform. (Note that the guarantee holds for continuous time gradient descent. It is technically possible, though not observed in practice, that SGD could produce a non-invertible kernel.) This remark was moved to the appendix due to lack of space but will be incorporated back into the main body. Additionally, the space of non-invertible kernels is measure zero in the space of kernels (it's rare for an eigenvalue to be *exactly* zero), and so non-invertible kernels are unlikely to occur by chance.

### Reviewer #3

Thank you for your comments and feedback. The expressivity/flexibility of the CONF is of a great deal of interest to us as well! In addition, we are very interested in better understanding the implicit bias over trained probabilistic models induced by this choice of architecture. We hope in future work to further explore these questions.

[Meta-Review · NeurIPS 2019]

This is a very good submission in the area of generative probabilistic models. The authors introduce normalizing flows using circular and symmetric convolutions leading to efficient Jacobian determinant computation. Several experiments have been conducted to demonstrate the model showing good results. Please revise the paper in light of the pre and post rebuttal comments of the reviewers. Congratulations on this excellent piece of work!